# Can Breastfeeding Protect from Multi-System Inflammatory Syndrome in Children (MIS-C)? A Preliminary Study

**DOI:** 10.3390/children9081160

**Published:** 2022-08-02

**Authors:** Xavier Rodríguez-Fanjul, Sergio Verd, Sonia Brio

**Affiliations:** 1Intensive Care Unit, Department of Paediatrics, Germans Trias i Pujol Hospital, Canyet Rd., 08916 Badalona, Spain; xavierrodriguezfanjul@gmail.com; 2Paediatric Unit, Department of Primary Care, Matamusinos Street, 07013 Palma de Mallorca, Spain; 3Balearic Institute of Medical Research (IdISBa), Valldemossa Rd., 07120 Palma de Mallorca, Spain; 4Intensive Care Unit, Department of Paediatrics, Santa Creu i Sant Pau Hospital, Mas Casanovas Street, 08041 Barcelona, Spain; sbrio@santpau.cat

**Keywords:** COVID-19, breastfeeding, Kawasaki disease, autoimmune diseases, infant formula

## Abstract

Background: Breastfeeding prevents Kawasaki disease (KD), as well as several autoimmune disorders. Since there is an overlap between the Multi-System Inflammatory Syndrome in children following SARS-CoV-2 infection (MIS-C) and KD, this case series aims to analyze the association between breastfeeding and MIS-C. Methods: A series of 16 cases of children with MIS-C admitted to three pediatric facilities between January 2021 and May 2022 were conducted. Breastfeeding rate was estimated through the Brief Breastfeeding and Milk Expression Recall Survey. Results: Out of 16 children, 9 (56%) had been breastfed at birth. Discussion: Our breastfeeding rate is below the median Spanish rate for initial breastfeeding. These findings do not clearly support the hypothesis that breastfeeding might prevent MIS-C. Conclusion: Contrary to the role of breastfeeding in KD prevention, our case series cannot answer with certainty the question about whether or not breastfeeding does protect children against MIS-C. These findings require confirmation in larger studies.

## 1. Introduction

In 2016, Japanese researchers observed that children who were breastfed were less likely to be hospitalized for Kawasaki disease (KD) compared with those who were fed formula (OR: 0.26; 95% CI: 0.12–0.55) [1]. Subsequent studies from Germany, Taiwan, or China were consistent with those findings.

Breastfeeding may confer protection against several autoimmune childhood diseases (multiple sclerosis, asthma, or inflammatory bowel disease) [2]. Multi-System Inflammatory Syndrome in children following SARS-CoV-2 infection (MIS-C) and KD are clinically similar diseases with different pathogenesis. However, the pathogenesis of the MIS-C has some overlapping features with KD suggestive of a likely autoimmune and autoinflammatory origin [3]. A cytokine storm plays an important role in MIS-C, with elevated levels of IL-1β, IL-6, IL-8, IL-10, and IL-17 [4]. This leads to the multi-organ involvement noted in MIS-C patients [5]. IL-6 and IL-17A levels have been found to be elevated in patients with KD compared to patients with MIS-C, suggesting a different pathogenesis between the two diseases [6]. However, similar to KD, immune complexes might lead to endothelial injury, and the response of MIS-C patients to intravenous immunoglobulins seems to support this thought [7]. Hence, we hypothesize that, just as is the case with KD, breastfeeding might have an important role in preventing children contracting an hyperinflammatory disease that culminates in MIS-C [4,8].

Since we have shown that breastfeeding was associated with a lower risk of positive SARS-CoV-2 RT-PCR test results in symptomatic children [9], the objective of this clinical study was to assess whether breastfeeding can also confer long-term benefits against a novel hyperinflammatory consequence associated with COVID-19 (MIS-C) years after infant feeding is terminated.

## 2. Methods

This is a retrospective series study that recruited all consecutive cases of MIS-C in children from January 2021 to May 2022 admitted to the pediatric intensive care units and pediatric wards of Sant Pau, Joan XXIII and Germans Trias University Hospitals. Informed consent was obtained from all parents’ patients. Sixteen parents of children with MIS-C were asked to participate in this research and all of them accepted to participate. Not one parent revoked her informed consent, with sixteen patients remaining in the study. Data were obtained from medical records and by phone survey, including: children’s age, birth weight, and timely initiation of breastfeeding. Estimation of breastfeeding rate: a brief interviewer-administered question was used to collect quality data recalled about lactation; this included the first modified questions of the validated Brief Breastfeeding and Milk Expression Recall Survey [10]. Sant Pau and Germans Trias hospitals handle all urgent and major medical cases from the Eastern part of Barcelona. Sant Pau and Germans Trias Hospitals admit roughly 2200 and 1300 children per year, and provide broad-reference clinical services for one million and eight hundred thousand people, respectively. Joan XXIII Hospital handles all urgent and major pediatric cases from the Southwest of Barcelona (Tarragona county) and provides reference clinical services for roughly 600,000 people.

## 3. Results

The study comprised sixteen consecutive children with MIS-C who attended our hospitals during the study period. The general features of the sixteen children included in the case series study are shown in Table 1. Nine of them (56%) had been breastfed. The median duration of any breastfeeding was 6 months (range: 1.5–24; interquartile range: 2.5–13).

There was no difference in demographical or clinical variables including birthweight, age, sex, or days of stay in intensive care or in the pediatric ward, between those who received any amount of breastfeeding after birth and those who did not (Table 2).

## 4. Discussion

The rate of breastfeeding in our case series is below current Spanish standards. According to the 2017 National Health Survey (ENSA) [11], 74% of 6-week-old infants are exclusively breastfed, as opposed to our figure of 56% of ever-breastfed children. In addition, it has been reported that eight out of ten Spanish children are breastfed at hospital discharge. Conversely, we found that only nine out of sixteen children with MIS-C from our case series had been breastfed at birth. To summarize, we show that Spanish children in the national sample have about a one-third-better rate of breastfeeding initiation than children with MIS-C in the current sample. For that reason, the protective role of breastfeeding against MIS-C is an idea that cannot be easily discarded and requires further assessment. Many limitations of our study stem from the limited data obtained at this stage, but similarly to KD [2,12], we report that breastfeeding does not make a difference regarding the short-term outcomes of MIS-C. In our sample, breastfed children were admitted to intensive care or to the pediatric ward for the same period of time that was medically necessary for formula-fed children. In the same way, most studies have found that breastfeeding confers protection against the development of KD, but have also found no effect or modest beneficial effects of breastfeeding on the clinical outcomes of children with KD [13].

Despite extensive research, the cause of KD remains unknown. However, coincidental aspects of KD and MIS-C [13] have reinforced the hypothesis of an airborne infectious trigger during KD. In the event that a lactating mother was exposed to the infection responsible for triggering KD, some markers of her immune response could have been shared with her baby through breast milk and would be able to modulate the immune response of the child in the long term. Conversely, this explanation does not apply to MIS-C since SARS-CoV-2 is a new virus, which renders it impossible that mothers of children with MIS-C were exposed to this virus in the past. Eventually, it may be the case that there was some cross-reaction of SARS-CoV-2 with other transmitted antibodies from breastfeeding.

We acknowledge that this is a very small sample, and these findings cannot answer with certainty the research question about whether or not formula-fed children are at higher risk for developing MIS-C compared with their breastfed counterparts.

## 5. Conclusions

Contrary to the role of breastfeeding in KD prevention, our experience shows that it remains uncertain whether breastfeeding affects MIS-C. These findings require confirmation in larger studies before they can assist in differentiating between KD and MIS-C.

## Figures and Tables

**Table 1 children-09-01160-t001:** General features of children with MIS-C.

Case	PICU/Ward	Gender	Age (Years)	Birthweight (Grams)	Initial Infant Feeding	Duration of Any Breastfeeding (Months)	Hospital Stay (Total Number of Days)	PICU Stay Length (Days)
1	GT	girl	7	3050	breastfeeding	1.5	6	2
2	GT	girl	6	3450	breastfeeding	3	5	0
3	GT	girl	5	2930	formula	0	6	0
4	GT	boy	4	3250	formula	0	8	2
5	SP	girl	7	3440	breastfeeding	14	6	0
6	SP	boy	4	3550	breastfeeding	24	8	3
7	SP	boy	2	2470	breastfeeding	2	5	0
8	SP	girl	4	3150	formula	0	3	0
9	SP	boy	4	3130	breastfeeding	3	6	0
10	J23	girl	4	3600	breastfeeding	6	4	0
11	J23	girl	7	3500	breastfeeding	12	6	1
12	J23	girl	5	3900	breastfeeding	8	4	0
13	J23	boy	3	2500	formula	0	7	3
14	J23	girl	5	3300	formula	0	5	0
15	J23	boy	3	3800	formula	0	8	2
16	j23	boy	4	2490	formula	0	6	0

Abbreviations: GT, Germans Trias Hospital; J23, Joan XXIII Hospital; MIS-C, Multi-System Inflammatory Syndrome in Children; PICU, Pediatric intensive care unit; SP, Sant Pau Hospital.

**Table 2 children-09-01160-t002:** Clinical characteristics of never and ever breastfed children.

Variables	Formula-Fed Children N = 7	Breastfed Children N = 9	*p* Value
Girls/boys	3/4	6/3	0.614
Birthweight (grams)	3060 (466)	3343 (412)	0.218
Years of age	4.00 (0.82)	5.11 (1.76)	0.147
PICU stay length (days)	1.00 (1.29)	0.67 (1.12)	0.588
Hospital stay (days)	6.14 (1.77)	5.56 (1.24)	0.447

Data are presented in mean (standard deviation). Abbreviations: N, number of participants; PICU, Pediatric intensive care unit.

## Data Availability

Data are available from the authors upon reasonable request.

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
