# Peer review of "Can Breastfeeding Protect from Multi-System Inflammatory Syndrome in Children (MIS-C)? A Preliminary Study"

_children, 2022, doi:10.3390/children9081160_

Round 1

Reviewer 1 Report

1. The significance of the difference between 74% and 63% is unclear. We can also conclude that children with MIS-C grown by breastfeeding are less than the general population of children grown by breastfeeding, because there may be some cross-reaction by other transmitted antibodies from breastfeeding. 

2. No description of the total number of MIS-C patients before getting informed consent for this study. Or do the following sentences mean that all patients are eight?

'Informed Consent was obtained from all parents’ patients. No one parent revoked her Informed Consent, with eight patients remaining in the study.

3. Anyway the number of patients is too small, so the title needs to include the word 'preliminary'.

Author Response

We are grateful for the insightful comments of Reviewer 1 regarding our paper. We have incorporated most of the suggestions made by him/her.

Question 1. The significance of the difference between 74% and 63% is unclear. We can also conclude that children with MIS-C grown by breastfeeding are less than the general population of children grown by breastfeeding, because there may be some cross-reaction by other transmitted antibodies from breastfeeding.
Answer: Eight new cases have been collected, hence the rate of breastfeeding in our enlarged sample has dropped from 63% to 56%. We agree to write that the difference between 74% and 56% is not clear, but a link between MIS-C and lack of breastfeeding cannot be completely dismissed. We have also added to the Discussion the possibility of cross-reaction by other transmitted antibodies from breastfeeding.
Question 2. No description of the total number of MIS-C patients before getting informed consent for this study. Or do the following sentences mean that all patients are eight?
'Informed Consent was obtained from all parents’ patients. No one parent revoked her Informed Consent, with eight patients remaining in the study.'
Answer: Absolutely all parents that were asked to participate in this research, actually have accepted to participate in it. We agree with reviewer 1 that the previous statement was confusing, and we have added this sentence to Methods.
3. Anyway the number of patients is too small, so the title needs to include the word 'preliminary'.
Answer: We have added the word preliminary to the title. Anyway, we have been fortunate to add eight new cases. New Table 1 includes 16 cases, and we have built Table 2 to summarize clinical outcomes of these patients.

Reviewer 2 Report

I believe that in this work there is a lot of confusion between Kawasaki disease and MISC, clinically similar diseases with completely different pathogenesis. It seems to me that, as the title proposes, suggesting a distinction between Kawasaki disease and a MISC through breastfeeding is too reductive. In addition, it is a case series of only 8 cases. I think the article could become more interesting by focusing only on breastfeeding and MISC, avoiding continuous comparisons with MK

Author Response

We appreciate the time that Reviewer 2 dedicated to providing feedback on our manuscript.

Question. I believe that in this work there is a lot of confusion between Kawasaki disease and MISC, clinically similar diseases with completely different pathogenesis. It seems to me that, as the title proposes, suggesting a distinction between Kawasaki disease and a MISC through breastfeeding is too reductive. In addition, it is a case series of only 8 cases. I think the article could become more interesting by focusing only on breastfeeding and MISC, avoiding continuous comparisons with MK.

Answer: In an attempt to clarify the distinction between KD and MIS-C, we have added a sentence to the Introduction on the clinical similarities and different pathogenesis of these conditions. We agree with Reviewer 2 that suggesting a distinction between KD and MIS-C through breastfeeding is too reductive. This is all the more true given that the rate of breastfeeding (54%) has become smaller as our sample has been enlarged. We have changed the title from “infant feeding might distinguish” to “infant feeding cannot distinguish”. We have collected eight new cases, our sample includes now 16 children with MIS-C from three Hospitals. We have built Table 2, that focuses on clinical outcomes of breastfed and formula-fed children. We conclude that we cannot answer our research question.

Round 2

Reviewer 1 Report

The authors well responded and revised their additional cases.

If there are 16 MIC-C, even though there is no significant difference,  quality is fairly good and persuasive.

Author Response

Dear Editor,

Thank you for giving us the opportunity to submit a revised draft of our manuscript for publication by Children.

We have done our best to respond to every point raised by the reviewers. 

REVIEWER 1

Comments and Suggestions for Authors

The authors well responded and revised their additional cases.

If there are 16 MIS-C, even though there is no significant difference, quality is fairly good and persuasive.

Answer: We appreciate the time that Reviewer 1 dedicated to providing feedback on our manuscript.

Reviewer 2 Report

I will change the title: can breast feeding protect  from misc? A preliminary study

Or something similar avoiding  comparison with kd

Line 28 these findings do not cleary support the hypotesis...

Line 48 50 some steps of the pathogenesis of misc and kd  have common point ....but you have to specificate  these steps and producing references

Line 58 do you mean  breast feeding is protettive from SarsCov2 infection and you would like to demonstrate if it is protettive from its consequences id est from misc ? 

Line 118 120 can you explain in a easier way?

Line 123 similarly to kd: can  you provide a reference?

Author Response

Dear Editor,

Thank you for giving us the opportunity to submit a revised draft of our manuscript for publication by Children.

We have done our best to respond to every point raised by the reviewers. 

REVIEWER 2

Comments and Suggestions for Authors:

I will change the title: can breast feeding protect from misc? A preliminary study

Or something similar avoiding comparison with kd

Answer: We agree, we have changed the title as suggested. The new title of the report seems to be the most appropriate one.

Line 28 these findings do not cleary support the hypotesis…

Answer: We have accepted the suggestion by Reviewer 2. This phrase has been changed as suggested.

Line 48 50 some steps of the pathogenesis of misc and kd have common point ....but you have to specificate these steps and producing references.

Answer: We have added a paragraph on the steps that suggest pathogenesis overlapping at some point, with a reference to the treatment that supports this thought.

Line 58 do you mean breast feeding is protective from SarsCov2 infection and you would like to demonstrate if it is protective from its consequences id est from misc ?

Answer: Yes, this is what we want to analyze. We have added a sentence to make it clear.

Line 118 120 can you explain in an easier way?

Answer: We have done our best to clarify this point, we have changed this sentence

Line 123 similarly to kd: can you provide a reference?

Answer: We refer now to a set of German children: patients with KD were breastfed for a shorter period of time, but breastfeeding duration did not have any effect on IVIG resistance [Meyer, 2019]; and to a case-control study of Chinese children: exclusive breastfeeding decreased the risk of developing KD, but was not associated with differences in risk of developing coronary artery lesions [Wang, 2020].

Answers Submission Date

 26th July, 2022

Round 3

Reviewer 2 Report

It sound well a conducted study